# Management and Treatment of Varicocele in Children and Adolescents: An Endocrinologic Perspective

**DOI:** 10.3390/jcm8091410

**Published:** 2019-09-08

**Authors:** Rossella Cannarella, Aldo E. Calogero, Rosita A. Condorelli, Filippo Giacone, Antonio Aversa, Sandro La Vignera

**Affiliations:** 1Department of Clinical and Experimental Medicine, University of Catania, 95125 Catania, Italy; 2Department of Experimental and Clinical Medicine, University Magna Graecia of Catanzaro, 88100 Catanzaro, Italy

**Keywords:** pediatric varicocele, testicular volume asymmetry, peak retrograde flow, varicocele repair

## Abstract

Pediatric varicocele has an overall prevalence of 15%, being more frequent as puberty begins. It can damage testicular function, interfering with Sertoli cell proliferation and hormone secretion, testicular growth and spermatogenesis. Proper management has a pivotal role for future fertility preservation. The aim of this review was to discuss the diagnosis, management and treatment of childhood and adolescent varicocele from an endocrinologic perspective, illustrating the current evidence of the European Society of Pediatric Urology (ESPU), the European Association of Urology (EAU), the American Urological Association (AUA) and the American Society for Reproductive Medicine (ASRM) scientific societies. According to the ASRM/ESPU/AUA practice committee, the treatment of adolescent varicocele is indicated in the case of decreased testicular volume or sperm abnormalities, while it is contraindicated in subclinical varicocele. The recent EAS/ESPU meta-analysis reports that moderate evidence exists on the benefits of varicocele treatment in children and adolescents in terms of testicular volume and sperm concentration increase. No specific phenotype in terms of testicular volume cut-off or peak retrograde flow (PRF) is indicated. Based on current evidence, we suggest that conservative management may be suggested in patients with PRF < 30 cm/s, testicular asymmetry < 10% and no evidence of sperm and hormonal abnormalities. In patients with 10–20% testicular volume asymmetry or 30 < PRF ≤ 38 cm/s or sperm abnormalities, careful follow-up may ensue. In the case of absent catch-up growth or sperm recovery, varicocele repair should be suggested. Finally, treatment can be proposed at the initial consultation in painful varicocele, testicular volume asymmetry ≥ 20%, PRF > 38 cm/s, infertility and failure of testicular development.

## 1. Introduction

Testicular varicocele is defined by the abnormal dilation and tortuosity of the pampiniform plexus draining the testes. The prevalence of varicocele is a debated issue since it depends on the selected population (infertile, fertile, age of patients) or the methods used to make the diagnosis (clinical examination and/or Doppler ultrasound). Most of the early epidemiological studies reported that the prevalence of varicocele in the general male adult population is approximately 15%, despite more recent studies suggesting the occurrence of an age-related prevalence or that it is inversely correlated with the body mass index. The prevalence of varicocele also seems to differ among fertile and infertile men or in those with primary or secondary infertility [1].

The prevalence in childhood and adolescence mirrors that in adulthood. Recently, a European study carried out in 7000 young men (median age: 19 years) reported the occurrence of varicocele in 15.7% of cases [2]. In a cohort of 4052 Turkish children and adolescents, its prevalence was 0.8% in 2–6-year-old boys, 1% in 7–10-year-old boys, 7.8 in 11–14-year-old boys, and 14.1% in 15–19-year-old boys [3], indicating an increasing prevalence as puberty begins.

Several data, mainly obtained from the adult population, suggest that varicocele has a negative role on testicular function. Accordingly, poorer semen quality and pregnancy outcomes have been reported in patients with varicocele compared to healthy controls [4,5,6]. In agreement, varicocele repair has been shown to improve both conventional (sperm concentration, progressive motility and normal forms) and bio-functional sperm parameters (percentage of spermatozoa with low mitochondrial membrane potential, phosphatidylserine externalization, abnormal chromatin compactness and DNA fragmentation) [4,6] and the outcome of assisted reproductive techniques (ARTs) [5]. In particular, the best quality of evidence comes from a meta-analysis provided by the American Society for Reproductive Medicine (ASRM) performed on 1241 patients with oligozoospermia or azoospermia and a history of varicocele undergoing ART. Pregnancy, live birth and sperm extraction rates were assessed following varicocele repair. Treatment led to improvement in pregnancy and live birth rates in oligozoospermic (odds ratio (OR) 1.699 and 2.366) and combined oligospermic/azoospermic patients (OR 1.761 and 1.760) compared to untreated patients. In addition, the live birth rate was higher in patients undergoing intra-uterine insemination (IUI) (OR 8.36). Similarly, the sperm retrieval rate increased after varicocele repair in azoospermic patients (OR 2.509). Based on such evidence, varicocele repair should be considered as a treatment option in oligozoospermic or azoospermic patients before they undergo ART [5]. Interestingly, the negative impact of varicocele on testicular function might also be extended to the Leydig cells. Indeed, according to meta-analytic data, its repair resulted in an increase in testosterone levels by 97.48 ng/dL [7].

The consequences of varicocele on testicular function in childhood and adolescence have been investigated to a lesser extent. In the last decade, a consensus was reached on the conditions requiring varicocele repair in adulthood [8]. On the contrary, several aspects of the management and treatment of varicocele in childhood and adolescence are poorly defined and still debated.

Adolescents with varicocele are highly heterogeneous, due to rapid changes in hormone levels and the stage of pubertal development [9]. This makes a standard approach more difficult. The current challenge is to establish which patient should be treated, when and what type of treatment should be preferred [10].

The aim of this review was to discuss the impact of varicocele on testicular function in childhood and adolescence from an endocrinological perspective and to highlight the best practice in diagnosis, management and treatment, according to the established guidelines by the European Society of Pediatric Urology (ESPU), European Association of Urology (EAU), the American Urological Association (AUA) and the ASRM.

To accomplish this, an extensive search in PubMed, Embase and Cochrane Library was performed by two independent authors (RC and RAC) using the following key-words: “varicocele”, “childhood”, “adolescent”, “diagnosis”, “management”, “treatment”, “sperm analysis”, “testicular volume”, “AMH”, “Inhibin B”, “nutcracker”. Only English language studies published from each database inception up to 30 July 2019 have been included. In addition, reference lists form articles were searched. No restriction on study design was used.

## 2. Pathogenesis of Testicular Damage

The evidence collected in children and adolescents suggests a negative impact of varicocele on testicular function, including sperm abnormalities, testicular hypotrophy and hormone alterations.

In more detail, varicocele is able to alter the conventional semen parameters in youth. A meta-analysis carried out in 357 patients with varicocele aged 15–24 years reported a statistically significant decrease in sperm concentration (−24 million/mL), motility (−7.5%) and morphology (−1.7%) compared to 427 age-matched controls [11]. Reflecting the findings in adulthood [4,5,6], its repair led to a significant improvement in sperm concentration and motility by 14.6 × 10^6^/mL (95% CI (7.1–22.1 10^6^/mL)) and 6.6% (95% CI (2.1–11.2%)), respectively, as shown by the same meta-analysis, suggesting a role of varicocele in the pathogenesis of sperm abnormalities [11].

Other data support these findings and indicate a role for varicocele-induced testicular hypotrophy in the establishment of sperm abnormalities. A study carried out in 57 Tanner stage V adolescent males at 14 to 20 years found that patients with testicular volume asymmetry greater than 10% had lower sperm concentrations and total motile sperm counts compared to those with asymmetry lower than 10%. A greater decrease was found in patients with a differential greater than 20% [12]. Precisely, total motile sperm counts of 64, 32 and 10 million in patients with 10%, 15% and 20% testicular asymmetry have been reported [12].

Large epidemiologic data on 7035 young men with a median age of 19 years showed that the sperm concentration halved in patients with grade III varicocele compared with healthy controls. Interestingly, patients with varicocele had higher serum levels of follicle-stimulating hormone (FSH), lower inhibin B and higher levels of luteinizing hormone (LH) compared to controls [2]. Accordingly, a study carried out in 31 pre-pubertal patients with left and nine with bilateral varicocele with an average age of 12.55 years found an increase in inhibin B levels 12 and 26 weeks after varicocele repair and a negative correlation between inhibin B and FSH serum levels [13]. Taken together, these findings suggest the occurrence of a Sertoli cell dysfunction in adolescent patients with varicocele, since inhibin B is secreted by these cells. This information is of particular interest for the assessment of testicular function in young patients, when sperm parameters cannot be evaluated.

The exact mechanisms by which varicocele can damage testicular function are not entirely clear. Several theories have been suggested so far.

Varicocele causes scrotal hyperthermia, which has a deleterious effect on spermatogenesis. Genetic factors play a role in the pathogenesis of varicocele, as it has a higher prevalence in first-degree relatives of men with known varicocele compared to controls [14] and also seems to be involved in the predisposition of varicocele-induced testicular damage. Accordingly, the genetically decreased expression of the heat-shock-proteins (HSPs) can contribute to heat stress, which, in turn, is associated with the markers of oxidative stress (OS) and apoptosis [15]. Transient exposure to high temperatures can reduce testis weight by interfering with spermatogenesis. In addition, blood stasis in varicose veins promotes leucocyte trapping, reactive oxygen species (ROS) overproduction and can cause testicular hypoxia. Low anti-apoptotic and increased pro-apoptotic gene expression (*HSP*, *Metallothireonin-1*, *BCL-2*, *BAX*, *PHUDA1*, *PRM2*, and *CCIN*) may confer susceptibility to OS and thermogenic damage, thus explaining the reason why high-degree varicocele does not always cause testicular damage [15].

Testes mainly consist of immature Sertoli cells which undergo proliferation and secrete the anti-Müllerian hormone (AMH) in childhood. In this phase, testicular volume reflects Sertoli cell proliferation. When puberty starts, Sertoli cells move from an immature to a mature state and lose the ability to proliferate. The final number of Sertoli cells reached at puberty will condition the spermatogenetic potential, since each Sertoli cell can support a defined number of germ cells [16].

Any event able to interfere with Sertoli cell proliferation in childhood may potentially impair testicular volume and the spermatogenetic potential in adolescence and adulthood. Importantly, a high temperature is able to affect Sertoli cell proliferation [17]. As a consequence, varicocele may hypothetically impact on the final Sertoli cell number in childhood (at least in some cases), thus inducing damage that cannot be reverted later in life. This highlights the importance of proper management and treatment in childhood/adolescence.

## 3. Evaluation

Adolescent varicocele is more frequently asymptomatic, despite chronic fullness or swelling in the scrotum or the inguinal area can be reported by the patient [10]. It is more frequently left sided, mainly due to the anatomical differences between left and right venous drainage. Testicular venous drainage enters the left renal vein at a sharp 90° angle. By contrast, it directly enters into the inferior vena cava in the right side. Also, the route of the left spermatic vein is longer than that of the right side. Furthermore, the left renal vein is compressed through the angle between the abdominal aorta and the superior mesenteric artery, when flowing into the inferior vena cava. These factors cause increased pressure in the left scrotum vein, leading to varicose veins occurrence [18]. A total of 3% are bilaterally palpable [10,19].

In most cases, adolescent varicocele is diagnosed during routine medical examination for school or sports or by testicular self-examination. Physical examination is the first step for the diagnosis of varicocele. It must be performed in the supine position for scrotum and genital inspection and for the palpation of testes, epididymis and deferent ducts. Testicular consistence must be appreciated, and the volume should be evaluated by Prader’s orchidometer, taking into consideration that it can overestimate the real value. The patient has to be asked to perform a Valsalva’s maneuver in the standing position. Varicocele usually presents as a plexus of veins with the consistency of a “bag of worms”. Clinical stadiation can be performed using the Dubin and Amelar system: grade 0 identifies subclinical varicocele (not clinically detectable, hence identified by ultrasound); grade I, which is palpable only during the Valsalva maneuver; grade II, appreciable also without the Valsalva maneuver; grade III, detected at inspection [20].

Scrotal ultrasound precisely evaluates testicular volume, the value of which is important when deciding whether varicocele repair is needed. It can be computed using the ellipsoid formula (length × width × thickness × 0.52), as described elsewhere [21]. Doppler ultrasound efficaciously indicates the varicocele grade, providing information on the maximum vein diameter and on the peak retrograde flow (PRF). In particular, varicocele is defined by the presence of multiple veins greater than 3.0–3.5 mm with concomitant retrograde blood flow. Reflux is classified as grade I (brief) when it lasts less than 1 s; grade II (intermediate) for 1–2 s and decreasing during the Valsalva maneuver and completely disappearing before the end of the maneuver; grade III (permanent) for reflux lasting more than 2 s and exhibiting a plateau aspect during the Valsalva maneuver [6]. Several scales are currently available to estimate the severity of varicocele. The most adopted ones are those developed by Sarteschi and by Dubin. The first, made of five different degrees, requires the patient to be lying down and standing. The Dubin scale includes three stages and is performed in the supine position [22]. These classifications are summarized in Table 1.

Patients exhibiting abnormally high venous velocity ratios should be evaluated for the nutcracker phenomenon (NcP) by Doppler ultrasound of renal vessels. The NcP is defined by compression of the left renal vein between the aorta and the mesentery artery. This causes renal venous hypertension and the dilatation of collateral veins, thus predisposing to varicocele. Recently, ultrasound criteria suggestive for the NcP have been proposed. These include a reduced aortic/superior mesenteric artery angle (normal values: 38–65 degrees), left renal vein compression at the origin of the aorta and superior mesenteric artery, increased flow velocity at the left renal vein, and left-sided varicocele with a vein lumen diameter > 3 mm [23]. Few studies have been developed on the NcP in youth. According to recent data, the NcP seems common among adolescents. In a cohort of 182 adolescents with clinical varicocele, the NcP was diagnosed in 77 patients (56.2%) who experienced higher velocity ratios than those without. In this cohort, the NcP has not been found to influence testicular asymmetry and initial or re-operative surgery [24].

Elastosonography is a non-invasive technique assessing testicular elasticity. It has already been used in undescended pediatric testes or adult varicocele. The prognostic value of elastosonography in adolescent varicocele has been recently investigated. Among a cohort of 30 patients with a clinically left varicocele, a significant change in testicular elasticity was found only in the case of volume asymmetry > 20% [25]. Currently, this technique is not routinely used in adolescents with varicocele.

Sperm analysis is an exam of pivotal importance. It has to be requested taking into account that the first ejaculation usually occurs 1.5 years after the onset of puberty [26]. Worryingly, it is not widely requested by pediatrics. A 2016 survey reported that only 13% of American pediatric urologists routinely evaluated sperm analysis in adolescent patients with varicocele. A total of 50% of them had some degree of discomfort in requesting this exam and discussing semen collection with patients so young [27]. This is alarming considering that several data confirm the negative impact of varicocele on sperm parameters in adolescence. Accordingly, a decreased total sperm count was found in 17–20-year-old patients with left varicocele and testicular ipsilateral hypotrophy [28]. Reduced sperm motility, vitality and morphology has also been reported in patients of 17–19 years with varicocele compared to age-matched controls [29]. Furthermore, sperm motility was profoundly more affected as basal blood flow velocity, maximal blood flow velocity, and pampiniform vein diameter increase [29]. In addition, a study carried out on 57 Tanner V patients (14–20 years old) reported lower sperm concentrations and total motile sperm counts in patients with a testicular differential > 10% compared to those with a differential < 10%. This was even more evident where the asymmetry > 20%, showing this group had a total motile sperm count < 10 million [12].

Despite the lack of consensus, the hormone profile may be useful for the workup of adolescent varicocele, since higher levels of FSH and LH and lower levels of inhibin B have been reported [2]. Other evidence confirmed the decrease in inhibin B levels, but not gonadotropin alteration [30]. AMH and inhibin B may be of particular utility when puberty is not already started and sperm analysis cannot be performed, especially when gonadotropin and testosterone levels are still not indicative. The testes have been considered silent in childhood for long. However, they secrete AMH and inhibin B in this phase. Particularly, serum AMH and inhibin B levels have already been suggested as markers of testicular function in pre-pubertal age [15]. Accordingly, impaired AMH and inhibin B levels have been reported in prepubertal and pubertal boys with varicocele [13,31].

## 4. Management

The management of childhood and adolescent varicocele is still controversial since no clear consensus has been reached so far. Spontaneous catch-up growth and sperm recovery has been observed in patients with varicocele, thus suggesting that its repair is not always necessary, but conservative management consisting of monitoring and follow-up can be suggested in selected cases. The challenge is to identify accurate predictive markers which can help to select those patients who will benefit from varicocele repair.

Conservative management has been suggested in Tanner V patients with no painful varicocele and normal testicular volume. A retrospective analysis of data from 216 patients with these clinical features showed a decreased total motile sperm count (< 20 million) in 45% of cases, with a spontaneous recovery in approximately 50% of patients with poor sperm parameters at baseline [32]. Overall, this indicates a lack of sperm recovery in 22.5% of cases with no painful varicocele and normal testicular volume. No additional marker was used in this study to characterize these patients [32].

Testicular asymmetry is one parameter that has been included in the decisional flowchart. Patients with values higher than 15%–20% have historically been treated with surgery. However, in 85% of adolescents with a > 15% testicular asymmetry catch-up, growth occurs without any intervention [33]. Hence, 2–3 testicular volume measurements at different follow-up times should be reasonably performed prior to deciding on varicocele repair [33], which may be suggested in the case of failure of testicular catch-up growth.

Another important parameter is the PRF, which has been suggested as a predictor of persistent or worsening testicular asymmetry in adolescent varicocele. After a 13.2 month-long follow-up, a study carried out in a cohort of 77 patients (age range: 9 to 20 years) revealed progressive asymmetry on follow-up examination in those with ≥ 20% asymmetry or PRF ≥ 38 cm/s. On this basis, patients presenting with these parameters should undergo varicocele repair after the initial consultation. On the contrary, those with PRF < 30 cm/s are less likely to require surgery and should be carefully monitored [34]. To raise the accuracy of predictive prognostic markers, a pilot study combining testicular volume asymmetry with PRF values showed that future worsening asymmetry was associated with ≥ 20% asymmetry and PRF > 38 cm/s (the so called “20/38 harbinger”). Accordingly, 94% of patients with the 20/38 harbinger did not have catch-up growth after a 15.5 month surveillance. Hence, intervention and not surveillance should be required in this set of patients [35,36]. In patients with borderline asymmetry or PRF, intervention has been suggested in the case of the abnormality of sperm parameters [37].

To summarize, “at-risk” patients deserving consideration for intervention are those presenting the following signs and symptoms [19]: (a) persistent abnormal sperm parameters with no evidence of recovery after surveillance; (b) pain; (c) persistently altered testicular volume asymmetry with a difference > 15–20% with no evidence of catch-up growth after surveillance; (d) PRF > 38 cm/s; (e) failure of testicular development; (f) 20/38 harbinger (which can also be considered to 15/38). Decreased AMH levels in children with varicocele may need careful surveillance due to the likely occurrence of Sertoli cell dysfunction. Prospective studies are needed to confirm this.

## 5. Treatment Options

Several treatment options are available for varicocele repair, including surgical (e.g., open inguinal-Ivanissevich, high retroperitoneal-Palomo, subinguinal, high inguinal, microsurgical-inguinal and subinguinal, laparoscopic) and radiological (sclerotherapy, embolization, antegrade vs. retrograde) approaches.

Overall, four meta-analyses have evaluated the effects of varicocele repair in childhood and adolescence so far [11,38,39,40]. Zhou and collaborators [38] reported an improvement in the bilateral testicular volume following varicocelectomy compared with observation, although no benefit on sperm parameters was found. By contrast, Nork and colleagues [11] observed a moderate improvement in sperm parameters. In greater detail, data collected from 357 patients with varicocele and 427 controls showed the presence of a significantly lower sperm concentration, motility and morphology. Studies where varicocele repair was performed by “Palomo” (open or laparoscopic) technique (*n* = 5), scleroembolization (*n* = 1), inguinal or sub-inguinal intervention with magnification (*n* = 4) were included to evaluate the effect of the treatment. Varicocele repair improved sperm concentration and motility. Each technique was effective in ameliorating the sperm outcome, despite scleroembolization only being used in a single study [11].

However, the above-mentioned meta-analyses [11,38] were biased by the inclusion of non-randomized comparative studies (NCTs), thus affecting the level of evidence. Locke and colleagues [39] compared testicular volume and sperm parameters in children and adolescents (up to 21 years old) with varicocele receiving surgical or radiological intervention with those receiving no treatment. By the inclusion of only randomized comparative studies (RCTs), they showed an improvement in testicular volume (mean difference 3.18 mL) and sperm count (mean difference 25.54 × 10^6^/mL) in treated patients compared with those undergoing conservative management. Overall, these data suggest the benefit of varicocele repair in childhood and adolescence. Nevertheless, this study [39] did not provide any information concerning a comparison of treatment options, surgical success, hydrocele formation, complication rates and paternity in the long term.

The best quality of evidence is offered by the latest systematic review and meta-analysis provided by the EAS/ESPU societies [40], including 12 RCTs (which were meta-analyzed), 47 NCTs (seven prospective and 40 retrospective) and 39 case series (which were qualitatively reviewed), for a total of 16,130 children and adolescents ≤ 21 years of age. The outcomes assessed were a short-term cure or success (evaluated < 9 months), testicular catch-up growth, pain resolution, sperm parameters and paternity (evaluated > 12 months) for benefits, complications such as testicular atrophy, hydrocele, wound infection, and failure rate for harms.

The success rate (disappearance of varicocele) was between 87% and 100% among RCTs. No difference was found neither between open and laparoscopic technique [41] nor between subinguinal and high inguinal varicocelectomy [42]. Similarly, the success rates were between 88.2% and 100% in the included NCTs and between 85.1% and 100% in the case series. Due to the lack of comparative data, no conclusion could be made concerning the type of treatment among these kinds of studies [40].

The available RCTs have assessed testicular catch-up growth in treated vs. untreated patients [43,44,45,46]. The majority of them compared inguinal and high inguinal varicocelectomy vs. observation. Only one RCT evaluated scleroembolization vs. observation [43]. Testicular volumes were significantly higher in the treated vs. untreated group (OR 1.52). NCTs report a testicular catch-up growth rate between 86 and 100% following embolization and between 62.8% and 97.1% following open varicocelectomy [40].

Data on sperm parameters coming from two RCTs [43,46] showed a significantly higher sperm concentration (mean difference 25.54 × 10^6^/mL) in treated vs. untreated groups, in the absence of any difference in sperm motility and morphology [40]. Overall, NCTs report an improvement in sperm parameters following surgical treatment, with a follow-up ranging between 17.6 months and 10.6 years. Similar data were reported in case series [40].

Although paternity rate is one of the most important outcomes, it is rarely reported due to the necessity of long-term follow-up. The study by Cayan and colleagues [47] assessed 286 patients and 122 controls. Patients were treated by microsurgical varicocelectomy. The paternity rate was 77.3% vs. 48.4% (treated vs. untreated) leading the authors to conclude the benefit of treatment in adolescent varicocele. By contrast, the study by Bogaert et al. [48], carried out in 661 boys (12 to 17 years old) with varicocele, showed no efficacy of sclerotherapy on change in paternity as adults. Accordingly, among the 361 respondents, 158 (43%) searched for paternity, which was achieved in 85% of the conservatively followed group and 78% of the active treatment group (*p* > 0.05).

The available evidence on the onset of harms coming from RCTs suggests a resolution of pain or recurrence of pain after treatment (laparoscopic varicocelectomy), despite only two RCTs mentioning this outcome [41,42], being diminished in up to 100% of patients [42]. Post-operative (both surgical and radiological) pain resolution rates reported in NCTs were 92.9% and 100% [40]. The most common complication reported was hydrocele, and atrophy, wound infection, hematomas, scrotal emphysema and shoulder pain were observed to a lesser extent. The rate of hydrocele formation 6–85 months post-varicocelectomy was 0–12%, being the lymphatic sparing surgery associated with a lower risk compared to the non-sparing one (OR 0.08) [40].

Generally, there is moderate evidence on the benefits of varicocele repair in children and adolescents, especially in those with high-grade varicocele, low left testicular volume, pain and poor sperm parameters. However, the superiority of a specific treatment approach cannot be identified [40]. Notably, while a radiological approach represents a valid technique, the long-term risk of radiation exposure in pediatric and adolescent population following percutaneous embolization procedure should be considered.

Finally, alternative strategies (e.g., anastomosis of the proximal part of the spermatic vein with the inferior epigastric vein) could be considered in the NcP [49].

## 6. Established Guidelines and Societies’ Positions

No guideline specifically deals with the management and treatment of childhood and adolescent varicocele. Current knowledge is extrapolated from guidelines endorsing the management of male infertility. Specifically, the ASRM/SMRU/AUA practice committee [50] suggests that varicocele diagnosis is made by th Dubin and Amelar clinical classification and to perform Doppler ultrasound only in case it is inconclusive. The treatment of adolescent varicocele is indicated in the case of decreased testicular volume or sperm abnormalities, while it is contraindicated in subclinical varicocele. The cut-off of testicular volume to suggest varicocele repair is not indicated and no specific treatment is recommended. Finally, these guidelines indicate to follow-up at least annually.

The EAU guidelines on male infertility [51] accordingly suggest using the Dubin and Amelar clinical grading classification and scrotal ultrasound to confirm the clinical findings. Worryingly, despite the authors stating that adolescent varicocele is often overtreated, no specific indication for the management and treatment of adolescent varicocele is provided. In contrast to the EAU guidelines, the EAS/ESPU meta-analysis [40] reported moderate evidence on the benefits of varicocele treatment in children and adolescents in terms of testicular volume and sperm concentration recovery. Accordingly, an ASRM society meta-analysis supports the efficacy of varicocele repair in youth on sperm concentration and motility [11].

## 7. Conclusions and Authors’ Recommendations

Varicocele evaluation has to be clinically performed using the Dubin and Amelar scale at first. Scrotal ultrasound should be requested to precisely define testicular asymmetry and PRF. Hormone detection (including AMH and inhibin B in childhood and LH, FSH and total testosterone in adolescence) should be carried out for a comprehensive evaluation of testicular function. Importantly, sperm analysis is of pivotal importance and it may be requested at least 1.5 years after the onset of puberty. Doppler ultrasound of renal vessels should be performed in selected cases (e.g., left-sided varicocele with a vein lumen diameter > 3 mm, hematuria, proteinuria, left-sided flank/lower abdominal pain, varicose veins, urinary frequency) [23].

Current evidence clearly indicates the impact of childhood and adolescent varicocele in testicular growth and sperm output. Spontaneous testicular catch-up growth can be observed in some cases. Some markers may be used to select patients who will benefit from varicocele repair. These mainly include testicular volume asymmetry and PRF. Therefore, we suggest that conservative management could be pursued in patients with PRF < 30 cm/s, testicular asymmetry < 10% and no evidence of sperm and hormonal abnormalities. In patients with 10–20% testicular volume asymmetry or 30 < PRF ≤ 38 cm/s or sperm abnormalities, careful follow-up is advisable. In the case of absent catch-up growth or sperm recovery, varicocele repair should be suggested. Finally, treatment can be proposed at the initial consultation in painful varicocele, testicular volume asymmetry ≥ 20%, PRF > 38 cm/s, infertility and failure of testicular development. On the basis of the current evidence, either radiological or surgical intervention may be prescribed (Figure 1).

## Figures and Tables

**Figure 1 jcm-08-01410-f001:**
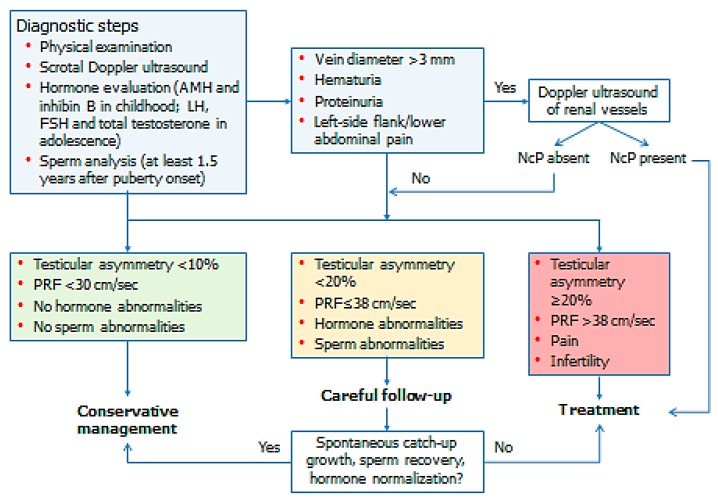
Management of childhood and adolescent varicocele.

**Table 1 jcm-08-01410-t001:** Ultrasound varicocele degree classifications.

Scale	Degree	Description
Sarteschi	I	Reflux detected only during the Valsalva maneuver, in the absence of evident scrotal varicosity during US study.
	II	Small posterior varicosity that extends to the superior pole of the testes. Their diameter increases and the reflux becomes detectable in the supratesticular region only during the Valsalva maneuver.
	III	Vessels appear enlarged in the superior pole only in the standing position. No enlargement can be detected in the supine position. Reflux is observed only during the Valsalva maneuver.
	IV	Vessels appear enlarged in the supine position. Dilatation is more marked during the Valsalva maneuver.
	V	Venus ectasia is detected in the prone and supine position. Reflux occurs at rest and it does not increase during the Valsalva maneuver.
Dubin	0	Moderate and transient venous reflux during the Valsalva maneuver.
	I	Persistent venous reflux that ends before the Valsalva maneuver is completed.
	II	Persistent venous reflux through the entire Valsalva maneuver.
	III	Venous reflux is basally detected and does not change during the Valsalva manuever

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
