# Peer review of "Management and Treatment of Varicocele in Children and Adolescents: An Endocrinologic Perspective"

_jcm, 2019, doi:10.3390/jcm8091410_

Round 1
Reviewer 1 Report
Report on Varicocle review paper
1. Abstract: the review is basically of the literature rather than the authors personal experience. Was there any systematic literature search? As the only special reference is to the three sets of guidelines, guide line conclusions should be included in the abstract.
2. Introduction: the Agarwal paper (their ref 1) concerns men with infertility. The prevalence figures given in this section are taken from the introduction of the Agarwal paper and are themselves a quote from a book on infertility. Contemporary primary sources are needed.
3. Introduction and pathogenesis: in both sections it is taken as given that varicocele is damaging to testicular function. Although there is an association between varicocele and diminished testicular function, there is no critical discussion of the evidence. For example, what is the prevalence of varicocele in men who have fathered children? What is the difference between statistical significance and clinical significance in the abnormalities found? As a really radical question – is it possible that varicocele and the testicular dysfunctions are all part of a syndrome of poor male development, varicocele being a component but not the primary cause?
4. Pathogenesis: assuming that varicocele does damage the testes, this section needs to be expanded and should make a more critical assessment of the evidence. With uncertainty about cause it is difficult to know whether the proposed treatments are likely to produce improvements.
5. Pathogenesis: There is no mention of the roles of exercise or of inheritance. There is literature on several aspects of these(1-3).
6. Pathogenesis: Their reference 13 is a good review article, but is mainly about experiments in small rodents. There are important data therein, but the present authors need to put it in context rather than just quoting it.
7. Evaluation: the authors state that the angle of entry of the left spermatic vein into the renal vein is the cause of the vastly increased prevalence of left varicocele compared to right. No evidence is given. Other veins drain into bigger vessels at quite sharp angles approaching 90o, such as the renal into the IVC, apparently without harm.
8. Evaluation: more work needs to be done on the semen analysis section. The authors put much weight on their reference 23. This paper is about oncology patients aged 18 – 65 yo who are about to be made infertile by cancer treatment. Patients were asked about their sexual milestones and it was found that 1st intercourse was a mean of 1.5 years after their self-reported age of puberty. It cannot be extrapolated from this that spermatogenesis would have reached its peak by then. Peak spermatogenesis may be two or more years after the end of apparent puberty. Spermiograms in the late teens can be misleading and improvements after varicocele ligation may just be due to increasing maturity.
9. Treatment options: this section needs to be sub-divided: surveillance, followed by the different surgical and radiographic options, with their advantages and disadvantages.
10. Outcomes: this would be a new section. What are the outcomes? It must be said that there are few good follow-up studies in the literature. Does surgery make the pain better? How many testes are lost due to surgery? Is the increase in testicular size due to more functioning tubules or to hypertrophy of existing ones or even to oedema? Are the improvements in testicular function clinically significant – for example, if a boy has a spermiogram worse than WHO norms, does surgery raise it above the norms; similarly, if it is above the norms before surgery, is a 10-15% improvement clinically important? Are there any paternity data? In this context, it would be important to discuss Bogaert et al 2013(4).
11. Established guidelines: guidelines are very important in maintaining good practice and they also influence medico-legal cases. Careful analysis thereof is essential in a review article. Silay et all – their reference 40 – does not 'show clear efficacy of varicocele surgery'. It states that 'there is moderate evidence of improvement in testicular volume and sperm concentration with absence of long-term outcomes, especially on fertility'. Reference 41 (Dong et al) is not a guideline. Reference 42 is guidelines for the male partners of infertile women and is irrelevant to the present paper. Reference 43, as the authors state, is about infertile males. Although it is possible to extract some data from adult guidelines, it must be done with extreme caution. It must be remembered that adolescents with varicocele are not ‘infertile males’ and, at present, there is little evidence that they will become so.
12. Conclusions: the authors must write their own conclusions providing they are based on the evidence that they have submitted in their review. I would only suggest that one of the conclusions would be to say whether they feel that varicocele in adolescents is actually harmful and whether correcting it makes them clinically better.
1. Rigano E, Santoro G, Impellizzeri P, Antonuccio P, Fugazzotto D, Bitto L, et al. Varicocele and sport in the adolescent age. Preliminary report on the effects of physical training. J Endocrinol Invest. 2004;27(2):130-2.
2. Scaramuzza A, Tavana R, Marchi A. Varicoceles in young soccer players. Lancet. 1996;348(9035):1180-1.
3. Gokce A, Davarci M, Yalcinkaya FR, Guven EO, Kaya YS, Helvaci MR, et al. Hereditary behavior of varicocele. J Androl. 2010;31(3):288-90.
4. Bogaert G, Orye C, De Win G. Pubertal screening and treatment for varicocele do not improve chance of paternity as adult. J Urol. 2013;189(6):2298-303.
Author Response
Comment 1: Abstract: the review is basically of the literature rather than the authors personal experience. Was there any systematic literature search? As the only special reference is to the three sets of guidelines, guide line conclusions should be included in the abstract.
Answer to comment 1: Details on literature search have been provided at the end of the introduction section (lines 83-88). As requested, we included guideline conclusions, followed by our recommendations, into the abstract (lines 19-31).
Comment 2: Introduction: The Agarwal paper (their ref 1) concerns men with infertility. The prevalence figures given in this section are taken from the introduction of the Agarwal paper and are themselves a quote from a book on infertility. Contemporary primary sources are needed.
Answer to comment 2: Prevalence of varicocele is a debated issue since it depends on the selected population (infertile, fertile, age of patients) or the methods used to make the diagnosis (clinical examination and/or Doppler ultrasound). Most of the early epidemiological studies reported that the prevalence of varicocele in the general male population is about 15%. More recent studies suggest an age-related prevalence or inversely correlated with the BMI. Finally, the prevalence seems to differ among fertile and infertile men, being the 25-30% and 50-80%, respectively. This evidence has been deeply discussed in a review focusing on the prevalence of varicocele, that we quote instead of Agarwal et al. (Asian J Androl 2016, 18(2), 179-81 – see reference n.1 of the revised paper). Accordingly, changes have been made in lines 37-43.
Comment 3: Introduction and pathogenesis: in both sections it is taken as given that varicocele is damaging to testicular function. Although there is an association between varicocele and diminished testicular function, there is no critical discussion of the evidence. For example, what is the prevalence of varicocele in men who have fathered children? What is the difference between statistical significance and clinical significance in the abnormalities found? As a really radical question – is it possible that varicocele and the testicular dysfunctions are all part of a syndrome of poor male development, varicocele being a component but not the primary cause?
Answer to comment 3: Thank you for your comment. The recent meta-analysis by Kirby et al., 2016 (ref. n. 5) published in Fertility and Sterility provides evidence from a causative association between varicocele and infertility. Accordingly, varicocele repair increased life birth rates and pregnancy rates in IVF and ICSI compared to not-treated patients which served as controls both in patients with oligozoospermia and in those with azoospermia. In particular, the OR of life birth rates in patients underground IUI after varicocele repair was as high as 8.36. This evidence justifies the causative role of varicocele in the pathogenesis of infertility and the suggestion to treat varicocele in patients with azoospermia or oligozoospermia undergoing to ART. In line with the evidence in the adulthood, another meta-analysis clearly showed the efficacy of childhood and adolescent varicocele repair (see reference n. 40).
Similar data on varicocele damaging effect on sperm parameters are provided by studies cites in references number 11 and 12 of the “Pathogenesis of testicular damage” section.
Furthermore, as stated by the World Health Organization, the prevalence of varicocele in fertile patients is significantly lower compared to infertile ones (11.7% vs 25.4%) (Fertil Steril. 1992 Jun; 57(6):1289-93).
Finally, despite it is an interesting hypothesis, there is currently no evidence that varicocele and poor testicular function belong to a syndrome of poor male development.
Comment 4: Pathogenesis: assuming that varicocele does damage the testes, this section needs to be expanded and should make a more critical assessment of the evidence. With uncertainty about because it is difficult to know whether the proposed treatments are likely to produce improvements.
Answer to comment 4: We appreciated this comment and this paragraph was accordingly extended. However, it has to be taken into account that the aim of this paragraph is to discuss the impact of childhood and adolescent varicocele on testicular function (not to provide evidence of the adult varicocele - a topic widely discussed in the literature). In addition, as replied in question number 3, the improvement in live birth and pregnancy rates in infertile patients undergoing to ART after varicocele repair, likely let hypothesize a pathogenic role of varicocele in male infertility. To make this clearer, we also expanded the Introduction section, in lines referring to the role of adult varicocele in testicular damage (see lines 51-68).
Comment 5: Pathogenesis: There is no mention of the roles of exercise or of inheritance. There is literature on several aspects of these (1-3).
Answer to comment 5: We have added the topic of inheritance quoting the reference n. 3 in lines 120-122 However, references 1-3 that you kindly suggested provided data on the role of inheritance and physical exercise on the pathogenesis of varicocele. Please consider that the primary aim of the paragraph “Pathogenesis of testicular damage” was to provide evidence on mechanisms by which varicocele has been shown to cause testicular damage and not on the pathogenesis of varicocele. Therefore, references n. 1 and 2 seem us out of focus.
Comment 6: Pathogenesis: Their reference 13 is a good review article, but is mainly about experiments in small rodents. There are important data therein, but the present authors need to put it in context rather than just quoting it.
Answer to comment 6: Reference number 13 reports evidence on the genetic predisposition to the varicocele-induced testicular damage, which is the primary focus of the paragraph entitled “Pathogenesis of testicular damage”. We added lines 121-122 to make it more clear.
Comment 7: Evaluation: the authors state that the angle of entry of the left spermatic vein into the renal vein is the cause of the vastly increased prevalence of left varicocele compared to right. No evidence is given. Other veins drain into bigger vessels at quite sharp angles approaching 90o, such as the renal into the IVC, apparently without harm.
Answer to comment 7: Thank you for your comment. There are anatomical differences between left and right renal venous drainage. The route of left spermatic vein is longer than the right side and the left spermatic vein vertically inserts into the left renal vein. These factors increase the pressure transmitted to the left scrotum vein and lead to varicose vein formation. Furthermore, the left renal vein is compressed when flowing into the inferior vena cava through the angle between the abdominal aorta and superior mesenteric artery, affecting the vein reflux, accompanied by the renal vein dilation.
The higher prevalence of left varicocele compared to the right is provided by the following study, which has been added to the reference list. The text has been accordingly modified (see lines 148-151).
Abdel-Meguid TA, Al-Sayyad A, Tayib A, Farsi HM. Does varicocele repair improve male infertility? An evidence-based perspective from a randomized, controlled trial. Eur Urol. 2011 Mar; 59(3):455-61.
Comment 8: Evaluation: more work needs to be done on the semen analysis section. The authors put much weight on their reference 23. This paper is about oncology patients aged 18 – 65 yr who are about to be made infertile by cancer treatment. Patients were asked about their sexual milestones and it was found that 1st intercourse was a mean of 1.5 years after their self-reported age of puberty. It cannot be extrapolated from this that spermatogenesis would have reached its peak by then. Peak spermatogenesis may be two or more years after the end of apparent puberty. Spermiograms in the late teens can be misleading and improvements after varicocele ligation may just be due to increasing maturity.
Answer to comment 8: The paper by Dabaja et al., 2014 is aimed to understand when to ask male adolescents to provide semen sample, due to the increasing need of fertility preservation in the adolescence in the field of oncology. To answer this question, the Authors prospectively investigated fifty males with no medical or sexual developmental abnormalities (they were not oncology patients). All subjects were asked four questions to self-report the age of the onset of puberty, when they first started to masturbate, their first experienced ejaculation and their first sexual intercourse. The first experienced ejaculation was 1.5 years after the onset of puberty in 80% present of the cohort and 84% started masturbation 1.5 years after the onset of puberty. This finding is also in line with previous evidence (Psychol Rep. 1996 Oct; 79(2):499-509; Pediatr Ann. 2005 Oct; 34(10):785-93).
Hence, the first sperm analysis should not be requested prior 1.5 years following the onset of puberty, simple because sperm collection is not possible. We did not report that spermatogenesis peaks at this time. For a proper management and counseling of childhood and adolescent varicocele, the knowledge of the timing of the first ejaculation is of importance. We have accordingly clarified this point (see lines 196-197).
Comment 9: Treatment options: this section needs to be sub-divided: surveillance, followed by the different surgical and radiographic options, with their advantages and disadvantages.
Answer to comment 9: According with your suggestion, the paragraph “Treatment options” was critically revised and restructured. We hope it now accomplished the Reviewer request (see lines 260-330).
Comment 10: Outcomes: this would be a new section. What are the outcomes? It must be said that there are few good follow-up studies in the literature. Does surgery make the pain better? How many testes are lost due to surgery? Is the increase in testicular size due to more functioning tubules or to hypertrophy of existing ones or even to oedema? Are the improvements in testicular function clinically significant – for example, if a boy has a spermiogram worse than WHO norms, does surgery raise it above the norms; similarly, if it is above the norms before surgery, is a 10-15% improvement clinically important? Are there any paternity data? In this context, it would be important to discuss Bogaert et al 2013(4).
Answer to comment 10: Outcomes have been included in the “Treatment options” section (see lines 288-330.). The paper by Bogaert et al 2013 has been discussed as requested (lines 313-316).
Comment 11: Established guidelines: guidelines are very important in maintaining good practice and they also influence medico-legal cases. Careful analysis thereof is essential in a review article. Silay et all – their reference 40 – does not 'show clear efficacy of varicocele surgery'. It states that 'there is moderate evidence of improvement in testicular volume and sperm concentration with absence of long-term outcomes, especially on fertility'. Reference 41 (Dong et al) is not a guideline. Reference 42 is guidelines for the male partners of infertile women and is irrelevant to the present paper. Reference 43, as the authors state, is about infertile males. Although it is possible to extract some data from adult guidelines, it must be done with extreme caution. It must be remembered that adolescents with varicocele are not ‘infertile males’ and, at present, there is little evidence that they will become so.
Answer to comment 11: Thank you for this comment.
Reference 41: you are right. We sought to refer to reference n. 11.
Concerning reference n. 40, lines 346-347 have been rephrased according with your considerations.
Although mainly endorsing the management of male infertility, the reference 42 specifically refers to the management of adolescent varicocele as following reported:
“Adolescent males who have unilateral or bilateral varicoceles and objective evidence of reduced testicular size ipsilateral to the varicocele may also be considered candidates for varicocele repair 6, 7, 8, 9. If objective evidence of reduced testis size is not present, then adolescents with varicoceles should be followed with annual objective measurements of testis size and/or semen analyses to detect the earliest sign of varicocele-related testicular injury. Varicocele repair may be offered on detection of testicular or semen abnormalities, as catch-up growth has been demonstrated as well as reversal of semen abnormalities; however, data are lacking regarding the impact on future fertility. Adolescents and young men not actively trying to conceive who have a varicocele and objective evidence of reduced ipsilateral testicular size may be offered varicocele repair”.
Reference 43 also refers to infertile patients and not to adolescents. We accordingly reported that no indication for management and treatment of adolescent varicocele is mentioned (lines 343-345).
Comment 12: Conclusions: the authors must write their own conclusions providing they are based on the evidence that they have submitted in their review. I would only suggest that one of the conclusions would be to say whether they feel that varicocele in adolescents is actually harmful and whether correcting it makes them clinically better.
Answer to comment 12: Our conclusions are stated in the last paragraph. We made them clearer (see line 362). As specified in the conclusive paragraph, varicocele repair should be suggested in selected cases (on the basis of testicular volume asymmetry, PRF, hormonal and sperm findings). A practical possible flow-chart is provided in Figure 1.
Reviewer 2 Report
This review article is well written and flows nicely and subsections were appropriate.
The authors discussed the subject of adolescent varicocele into an adequate depth
The references used were relevant to the message of the review article
There are some minor grammar/ text editing corrections required; for example, line 137; A half should be changed to Half.
Overall, this review article is well written. The subject/message is relevant to current management of varicocele and will be a useful reference paper for any future work.
Author Response
Comment 1: This review article is well written and flows nicely and subsections were appropriate. The authors discussed the subject of adolescent varicocele into an adequate depth. The references used were relevant to the message of the review article. There are some minor grammar/ text editing corrections required; for example, line 137; A half should be changed to Half.
Answer to comment 1: We are grateful for the time spent to reviewing the present paper. The revised manuscript has been spell-checked.
Reviewer 3 Report
The paper report the impact of varicocele on testicular function in childhood and adolescence from an endocrinological perspective. The paper is well written and presents certain interest in the field of varicocele and adolescence and may also be useful for counselling purposes. The authors should explain better the potential complications of surgical varicocele repair, based on the approach used. The Conclusions and Authors’ recommendations are critically discussed.
Author Response
Comment 1: The paper reports the impact of varicocele on testicular function in childhood and adolescence from an endocrinological perspective. The paper is well written and presents certain interest in the field of varicocele and adolescence and may also be useful for counselling purposes. The authors should explain better the potential complications of surgical varicocele repair, based on the approach used. The Conclusions and Authors’ recommendations are critically discussed.
Answer to comment 1: According with your suggestion (and that of the Reviewer 1), the paragraph “Treatment options” was critically revised and restructured. We hope it now accomplished the Reviewer request (see lines 260-330).
Round 2
Reviewer 1 Report
Thank you for asking me to review this re-submission.
Firstly, I apologise for the mistake that I made in my remarks on the Dabaja paper. I agree that the subjects were not oncology patients, though my other remarks about it stand.
The authors have made comprehensive alterations to the review and I believe that it is much improved. My original comments were aggressive but, I hope, challenged the authors in a way that a critical reader would do. I am sure that the authors were distressed by them, though they may feel that they were broadly helpful. The new version is much improved and recognises the uncertainties in the significance and management of varicocoele in adolescents.